# Single Exon Skipping Can Address a Multi-Exon Duplication in the Dystrophin Gene

**DOI:** 10.3390/ijms21124511

**Published:** 2020-06-25

**Authors:** Kane Greer, Russell Johnsen, Yoram Nevo, Yakov Fellig, Susan Fletcher, Steve D. Wilton

**Affiliations:** 1Centre for Molecular Medicine and Innovative Therapeutics, Murdoch University, Murdoch 6150, Australia; K.Greer@murdoch.edu.au (K.G.); R.johnsen@murdoch.edu.au (R.J.); s.fletcher@murdoch.edu.au (S.F.); 2Perron Institute for Neurological and Translational Science, Perth 6009, Australia; 3Institute of Neurology, Schneider Children’s Medical Center of Israel, Tel-Aviv University, Tel-Aviv 62919, Israel; yoramne@clalit.org.il; 4Pathology Department, Hadassah-Hebrew-University Medical Center, Jerusalem 91120, Israel; Fellig@hadassah.org.il; 5Centre for Neuromuscular & Neurological Disorders, University of Western Australia, Perth 6009, Australia

**Keywords:** Duchenne muscular dystrophy, dystrophin, antisense oligomers, duplication mutations, exon skipping, splicing

## Abstract

Duchenne muscular dystrophy (DMD) is a severe muscle wasting disease typically caused by protein-truncating mutations that preclude synthesis of a functional dystrophin. Exonic deletions are the most common type of *DMD* lesion, however, whole exon duplications account for between 10–15% of all reported mutations. Here, we describe in vitro evaluation of antisense oligonucleotide-induced splice switching strategies to re-frame the transcript disrupted by a multi-exon duplication within the *DMD* gene. Phosphorodiamidate morpholino oligomers and phosphorodiamidate morpholino oligomers coupled to a cell penetrating peptide were evaluated in a Duchenne muscular dystrophy patient cell strain carrying an exon 14–17 duplication. Two strategies were employed; the conventional approach was to remove both copies of exon 17 in addition to exon 18, and the second strategy was to remove only the first copy of exon 17. Both approaches result in a larger than normal but in-frame *DMD* transcript, but surprisingly, the removal of only the first exon 17 appeared to be more efficient in restoring dystrophin, as determined using western blotting. The emergence of a normal sized *DMD* mRNA transcript that was not apparent in untreated samples may have arisen from back splicing and could also account for some of the dystrophin protein being produced.

## 1. Introduction

Duchenne muscular dystrophy (DMD), an X-linked recessive condition, is the most common and severe form of childhood muscle wasting. DMD is generally caused by protein truncating mutations in the *DMD* gene and affects approximately 1:5200 live male births [1]. Those affected typically become wheel chair dependant by the age of 12 years, followed by death in the third decade of life due to cardiac and/or respiratory complications [2,3,4]. Mutations that lead to premature termination of translation have been found throughout all 79 exons of the 2.4 Mb *DMD* gene, with the most common type being genomic deletions of one or more exons that induce a frame-shift and disrupt the protein coding sequence [5]. Loss of the crucial 427 kD muscle-specific dystrophin isoform that normally links the myofibre sarcolemma to cytoskeletal actin compromises muscle fibre integrity during contraction and relaxation [6].

A milder allelic disorder, Becker muscular dystrophy (BMD), also arises from mutations in the *DMD* gene, most commonly in-frame deletions that allow synthesis of an internally truncated protein that retains some function. BMD patients generally exhibit a later age of onset, slower clinical progression and a longer lifespan. The BMD phenotype can be highly variable, depending upon the nature of the mutation, varying from borderline DMD (intermediate) to some cases where the reported phenotype is so mild that diagnosis was only made by chance [7]. The differences in disease progression between DMD and BMD form the basis of potential therapeutic interventions using antisense oligomers (AOs) that aim to induce exon skipping and generate a BMD-like dystrophin isoform from a *DMD*-mutated dystrophin gene [8,9,10].

Induced exon skipping is based on the observation that excising one or more exons from the *DMD*-mutated mature transcript can remove or by-pass mutations that cause premature termination of translation, allowing synthesis of an internally truncated semi-functional BMD-like dystrophin isoform. Those exon skipping studies in the clinic and in development describe restoration of the reading frame to address dystrophin genomic deletions, despite the fact that all early proof-of-concept and pre-clinical animal studies focussed on single exon excision to remove the mutated exon 23 in the mdx mouse dystrophinopathy model [11,12].

Human dystrophin exon duplications have received less attention, despite the fact these mutations potentially offer some advantages when considering strategies to restore the dystrophin reading frame [8,13]. Hypothetically, for some duplications this could be achieved by removing only the frame shifting or duplicated exons, producing dystrophin of a near normal length and structure. Here, we report two strategies aimed at restoring the dystrophin reading frame in vitro in patient cells disrupted by a *DMD* exon 14–17 frame-shifting duplication. Phosphorodiamidate morpholino oligomers (PMOs) and phosphorodiamidate morpholino oligomers coupled to a cell penetrating peptide (PPMOs) were used to target exons 17 and 18, or only exon 17. DMD-causing dystrophin exon duplications account for 10–15% of all cases, and similar to some DMD-causing non-duplication dystrophin mutations, multi-exon skipping should be required to restore the reading frame around most dystrophin duplications. However, in this particular *DMD* mutation arising from an exon 14–17 duplication the most efficient strategy was unexpectedly found to be the simplest: removal of only the first copy of exon 17. We compared this simple option to the conventional approach of restoring a semi-functional dystrophin by removing both copies of exon 17 in addition to exon 18 [14] and found the simplest to be the most efficient as shown by restoration of a near full-length protein.

## 2. Results

The dystrophin mutation in a myogenic dystrophic cell strain carrying a duplication of exons 14–17 was confirmed using RT-PCR. This duplication disrupts the reading frame and is consistent with the clinical diagnosis of DMD. As shown in Figure 1, this particular mutation would have the dystrophin reading frame restored by the excision of the first exon 17 only, or more conventionally, the removal of both copies of exon 17 and exon 18. 

The nucleotide sequences of dystrophin exons 17 and 18, along with 25 bases of the flanking intron, were interrogated using ESE Finder 3.0 to predict motifs involved in processing the dystrophin pre-mRNA, as shown previously [15]. AOs were designed to anneal to the known splice sites and predicted splice enhancer motifs across the exons and flanking intronic sequences. Oligomer sequences were first evaluated as individual AOs composed of 2′-O-methyl modified bases (2′OMe) on a phosphorothioate backbone, after transfection as cationic lipoplexes into unaffected human myogenic cells, as previously reported [16]. 

The most effective 2′OMe AOs targeting exons 17 and 18 (Table 1) were either transfected individually or combined in equimolar amounts and transfected into unaffected human myogenic cells to assess exclusion of the targeted exons [15]. The AO sequences targeting individual exons 17 and 18, optimized as 2′OMe oligonucleotides, were subsequently synthesised as PMO or PPMO, respectively. PMO H17A (+61+86) and PPMO H18A (+24+53) targeting exons 17 and 18 were combined in equimolar amounts and again tested in myogenic cells derived from a non-dystrophic individual, with some excision of both targets observed (Figure 2a). The PMO H17A (+61+86) targeting exon 17 (Figure 2b) was found to be more efficient than H17A (+10+35) (data not shown), which was designed to anneal to motifs closer to the exon 17 acceptor splice site.

Equimolar amounts of the PMO targeting dystrophin exon 17 (H17A (+61+86)) and PPMO targeting exon 18 (H18A (+24+53)) were transfected into myogenic cells derived from a DMD patient carrying an exon 14–17 duplication in *DMD* and left for 96 h before RNA and protein were extracted for analysis. The splice switching treatments included a combination of PMO and PPMO targeting exons 17 and 18, individual PMOs targeting exon 17 (H17A (+61+86) or H17A (+10+35)), a PPMO targeting exon 18 (H18A (+24+53)), a scrambled sequence control PMO and PMO H17A (+61+86) combined with a peptide designed to penetrate cells (Endo-porter™). After the treatment targeting the removal of both exon 17 and 18, we observed predominantly exon skipping of exon 18 after transfection at concentrations as low as 200 nM (100 nM of each PMO/PPMO). Individual PMOs targeting exon 17 (H17A (+61+86) and H17A (+10+35)) resulted in exon 17 skipping after transfection at concentrations as low as 200 nM for both treatments. Interestingly, an increase in the level of a normal sized product were detected in both exon 17 PMO transfections when compared to exon 18 treated, sham treated, Endo-porter™ delivered H17A (+61+86) or untreated control. The individual PPMO targeting exon 18 resulted in skipping of that target after transfection at concentrations as low as 200 nM. However, when targeting exon 18 with the PPMO we observed substantial cell death at 5 µM, making western blot analysis difficult. This in vitro toxicity can be attributed to the conjugated peptide tag on the PPMO, as previously reported [17,18]. As anticipated, the scrambled sequence PMO control did not induce any exon 17 or 18 skipping. Exon 17 (H17A (+61+86)) in combination with Endo-porter™ resulted in exon 17 skipping after transfection at the concentration of 5 µM (Figure 2c).

Long range and transcript specific RT-PCRs were performed on RNA from the transfected cells to distinguish which of the duplicated copies of exon 17 were being removed from the mature gene transcript. A forward primer in exon 16, when combined with a reverse primer in exon 15, would generate an amplicon (exon 16:17:15) from only the first (proximal) exon 17 of the duplication. We observed an increase in the proximal exon 17 excision in a dose dependant manner after transfection with either PMO H17A (+61+86) or H17A (+10+35) (Figure 3a). When H17A (+61+86) was combined with Endo-porter™, there was a clear signal above the background, which was present in all samples including from the sample with normal dystrophin gene expression. RNA from un-transfected myogenic cells derived from a healthy non-dystrophic individual also produced an amplicon of the size expected from this exon arrangement. However, high sensitivity amplification conditions were employed and the 472 and 296 bp products were only present in very low amounts in this extract from healthy cells, suggesting the presence of circular RNA species that have arisen from back splicing [19,20] (Figure 3a).

The same forward PCR primer in exon 16, when used in conjunction with a reverse primer in exon 19, should generate a more complex pattern with generation of amplicons spanning the latter portion of the duplication (exons 16:17:14:15:16:17:18:19 amplicon 1084 bp), as well as the normal dystrophin exons (16:17:18:19 amplicon 518 bp). The cells transfected with PMOs targeting both exon 17 and 18, or the exon 18 PMO (H18A (+24+53)) showed robust exon 18 excision, which obviously could only occur in the latter (normal) part of the *DMD* transcript. Exon 17 excision from the normal dystrophin transcript was evident but not as pronounced as exon 18 skipping, after transfection with H17A (+61+86), H17A (+10+35) and the Endo-porter™ treatment. As expected, there was no exon excision in the untreated or the scrambled sequence PMO treated cells (Figure 3b). The primer location and expected amplicon sizes for this PCR analysis is shown (Figure 3a,b). DNA sequencing was performed to confirm exonic arrangements in this mutation and the removal of the first exon 17 (Figure 4a,b).

Western blotting was undertaken on the protein extracts from these transfections, and DMD patient cells treated with the cocktail of H17A (+61+86) PMO and the exon 18 PPMO showed no increase in dystrophin compared to the baseline observed in untreated cells. Treatment with only the exon 17 PMO (H17A (+61+86)) showed an increase in dystrophin with approximately a 6-fold increase when compared to baseline untreated at 5 µM, 3-fold at 800 nM and 1.8-fold at 200 nM. Surprisingly, treatment with only exon 17 PMO (H17A (+10+35)) showed an increase in dystrophin of approximately 14-fold compared to baseline untreated at 5 µM, 4.5-fold at 800 nM and 1.8-fold at 200 nM. In contrast, the exon 18 PPMO-induced exon skipping appeared to ablate all dystrophin expression and the scrambled sequence control PMO at 5 µM had no effect on dystrophin levels relative to baseline untreated. Treatment with exon 17 PMO (H17A (+61+86)) in conjunction with Endo-porter™ at 5 µM showed a 20-fold increase in dystrophin when compared to baseline untreated. Baseline patient untreated dystrophin levels are approximately 0.1% of what is considered an “average” level in healthy non-dystrophic individuals (Figure 5).

To characterise the unique intronic breakpoint that would arise from this genomic rearrangement, PCRs spanning the intron 17:13 junction using numerous combinations of forward and reverse primers (Table 2) were performed on DNA from the patient with the duplication of exon 14–17 and DNA from the healthy control (Figure 6a). PCR products of approximately 3–8 kb from the exon 14–17 duplication patient were identified with 3 primer sets (63+76, 63+57 and 56+76). There was no product band of the same size generated from the healthy control sample (Figure 6b). Sanger sequencing of the PCR products confirmed the position of the intron 17: intron 13 breakpoint to be 1057 bases into intron 17 and 8157 bases upstream from exon 14. Two bases (ac) at the breakpoint junction could come from either intron 17 or intron 13 (Figure 6c).

## 3. Discussion

Dystrophin exon skipping is emerging as a promising therapeutic strategy for some DMD-causing gene lesions, with two compounds, *Eteplirsen* (for exon 51 skipping) and *Golodirsen* (for exon 53 skipping), granted accelerated approval by the US Food and Drug Administration and *Casimersen* in clinical trials for exon 45 skipping (NCT03532542). These exon skipping strategies are designed to restore the reading frame of dystrophin gene transcripts disrupted by common subtypes of frame-shifting deletions. Some DMD-causing mutations will not be amenable to targeted exon skipping, including massive deletions in excess of ~34 exons, or those mutations in exons encoding crucial domains. Nevertheless, we have designed a panel of splice switching compounds for every dystrophin exon, excluding the first and last. All *DMD* exons can be excised from the mature gene transcript, although there is considerable individual variation in the efficiency of exon removal [16].

While *DMD* exonic deletions are responsible for approximately 65% of DMD cases, dystrophin exon duplications are responsible for an estimated 10–15% of cases [14], with exon 2 being the most commonly duplicated exon. Nevertheless, reports of AO-induced exon excision to address duplications have to date been limited, and we are aware of only a handful of studies investigating exon skipping to overcome duplications in *DMD* [8,13,15,21]. We have previously described the correction of a dystrophin exon 18 duplication by excising both copies of exon 18, as well as exon 17 [15]. We have also investigated exon skipping in cells from two patients with exon 2 duplications in vitro, by removing only a single copy of exon 2 after dose-titration studies, or by removing an exon ‘block’ of exons 2–7 [13]. Aartsma-Rus and colleagues have reported successful skipping of duplicated exons 44 and 45, but did not achieve skipping of a larger duplication involving exons 52 to 62 [8]. A mouse model with a *Dmd* exon 2 duplication has been developed that should allow for duplication studies to be undertaken in vivo [21].

Here, we sought to address a frame-shifting duplication of exons 14–17 where the simplest strategy would be to remove only the first exon 17 in the duplication, thereby resulting in an in-frame dystrophin transcript that encodes a slightly larger but presumably functional protein. Although we have previously demonstrated the removal of one exon from an exon 2 duplication was feasible in vitro, by inducing low levels of skipping where the first or second duplicated exon is removed [13], such a strategy could be more challenging when dealing with duplications involving multiple exons. The possible exon combinations arising from skipping only one of the frame-shifting duplicated exons increases with the size and complexity of the mutation. The most logical approach for multi-exon duplications would be skipping a block to delete all duplicated exons responsible for the frameshift, plus additional flanking exons to restore the reading frame. Although this is possible, a multi-exon skipping approach is often inefficient, technically challenging [13] and would result in a shorter BMD-like dystrophin isoform [22]. Depending upon the location of the mutation, exon skipping strategies that employ the minimum number of splice switching oligomers and allow production of a near-normal length protein must be considered preferable and hence more likely to be clinically feasible.

However, in this particular duplication, relatively high concentrations of splice switching oligomers were needed to induce robust skipping of exon 18 in patient-derived cells, with only moderate skipping of exon 17 detected using RT-PCR. Consequently, it was somewhat surprising and initially counter-intuitive when we were able to demonstrate dystrophin expression using western blotting in treated patient cells after inducing exon 17 skipping with a single PMO targeting that exon. Furthermore, the in vitro restoration of some dystrophin was achieved at lower concentrations than that used in the conventional strategy targeting both exons.

In hindsight, it may not be surprising that the first duplicated exon 17 is excised in preference to the second or downstream exon 17. The first exon 17 in the duplication will be followed by an exon 14 and hence will be out of the normal context, whereas the next exon 17 will be preceded by exon 16 and followed by exon 18. Each different family *DMD* deletion or duplication will almost certainly have different intronic breakpoints, and this could potentially have a major influence on transcript processing [23]. In this patient, where ~21 kb of intron 17 and ~16 kb of intron 13 has been lost, the nature and position of an intron 17: intron 13 breakpoint and the exon 17 donor splice site interacting with the acceptor splice site of exon 14 could substantially weaken splicing of the out-of-context exon, compared to the second exon 17 being flanked by the normal exons 16 and 18. Unless another patient with the same duplication of exons 14–17 was found, these intron: intron interactions can only be hypothesised.

In addition to the moderate exon 17 skipping in the samples transfected with only the PMO targeting exon 17, there is also the appearance of a normal sized amplicon that was not present in untreated or sham treated patient samples. If this is not a PCR artefact arising from slippage of strands during the extension stage [24], it could perhaps indicate some back-splicing event that removes the duplicated exons, and further studies are warranted to explore this possibility. The use of RNAse R to degrade linear RNA [25,26], leaving circular RNA intact, may indicate whether treatment with the PMO targeting exon 17 induced significant levels of back splicing. The consistent generation of low levels of amplicons from the 16F and 15R RT-PCR from a normal individual indicates the presence of traces of circular RNAs, clearly indicating some back splicing is occurring in this region. The ladder of bands would indicate the presence of multiple species of circular RNAs from natural back splicing and exon skipping and could account for some of the low levels of dystrophin in the untreated patient cells. Nevertheless, single exon 17 skipping clearly resulted in many fold increases from baseline. Since the induction of single exon skipping and back splicing are not mutually exclusive, the combination of both may offer an explanation as to why the dystrophin protein levels have been restored to levels not anticipated.

In summary, in developing antisense strategies to overcome some multi-exon duplications in the *DMD* gene, it is evident that targeting only a single exon may be the most effective and therefore clinically viable option. Combinations of exon skipping compounds can induce dual or multi-exon skipping in vitro [13,15], but this has not yet come to the clinic. Logistically, it would be simpler to administer a single compound and presumably there would be fewer safety issues and less potential for off-target effects. It remains to be seen if this trend will be applicable to other multi-exon duplications and this can only be evaluated in patient-derived cells on a case-by-case basis. It is conceivable that other *DMD* exon 14–17 duplications may not behave in the same manner, since unrelated cases will almost certainly have different breakpoints. While developing strategies to address the common deletion subtypes, we must not forget that this is a personalized therapeutic intervention tailored to an individual mutation, and those with rare or challenging dystrophin rearrangements must not be overlooked.

## 4. Materials and Methods

### 4.1. AO Design and Synthesis

Splice switching AOs were designed to anneal to splicing motifs at the intron: exon boundaries, as well as ESE motifs predicted by the web-based application, ESEfinder [27]. AOs were first prepared as 2′OMe oligonucleotides on an Expedite 8909 Nucleic acid synthesiser using the 1 μM thioate synthesis protocol. PPMOs were supplied by Sarepta Therapeutics (Cambridge, MA, USA) and PMOs were supplied by Genetools, LLC (Philomath, OR, USA). Oligomer nomenclature is based on that described by Mann et al. [9] and indicates oligomer annealing coordinates.

### 4.2. Cell Propagation

Normal primary human myoblasts were cultured from biopsies taken from healthy individuals during elective surgery at Royal Perth Hospital, Perth, Western Australia. Informed consent was obtained, and the project was approved by the Human Ethics Committee of Murdoch University (approval number, 2013/156, 25 October 2013). Primary human myogenic cells were prepared as described by Rando and Blau [28] with minor modifications [29] and grown in Hams F10 Medium (Thermofisher Scientific, Melbourne, Australia) supplemented with foetal bovine serum (Thermofisher Scientific, Melbourne, Australia) and chick embryo extract (Jomar Life Research, Melbourne, Australia). Primary patient myoblasts were derived from a biopsy taken, after informed consent, from a DMD patient with a *DMD* duplication of exons 14–17. These were provided by the Simon Winter Institute of Human Genetics, Israel.

Prior to transfection, myoblasts were allowed to differentiate in 5% horse serum (Thermofisher Scientific, Melbourne, Australia) in Dulbecco’s modified Eagle medium (DMEM) (Thermofisher Scientific, Melbourne, Australia). Seeding occurred in pre-coated 25 cm^2^ flasks 3–4 days before transfection [29].

### 4.3. Transfection

Myogenic cells were transfected with PMO and/or PPMO in Opti-MEM (Thermofisher Scientific, Melbourne, Australia) at concentrations indicated. Transfection medium was removed after approximately 96 h.

### 4.4. RNA Extraction and RT-PCR Assays

A MagMax™-96 total RNA isolation kit (Thermofisher Scientific, Melbourne, Australia) was used to extract total RNA from cultured cells, according to the manufacturer’s guidelines. RT-PCR primers were designed to include several exons either side of the target area in order to minimise preferential bias of short (skipped) products during amplification. This assay also allows abnormal RNA processing, such as cryptic splice site activation or removal of flanking non-targeted exons to be detected.

Two PCR systems were used to evaluate the RNA. The first used approximately 50 ng of total RNA with Superscript III One-Step RT-PCR System with Platinum^®^ Taq DNA Polymerase (ThermoFisher Scientific, Melbourne, Australia). Conditions were as described in Wilton et al. [15].

The second method of RT-PCR involved the conversion of total RNA to cDNA using SuperScript IV Reverse Transcriptase (ThermoFisher Scientific, Melbourne, Australia) with a gene specific reverse primer in *DMD* exon 25. All other conditions were performed according to the manufacturer’s instructions (ThermoFisher Scientific, Melbourne, Australia). PCR (35 cycles) was performed on the equivalent of 50 ng of the original RNA template using TaKaRa LA Taq polymerase (TAKARA Biotechnology (Dalian) Co., LTD., Shiga, Japan) with LA amplification buffer, according to the manufacturer’s instructions [15]. An additional 25 cycles were performed using 1 µL of template from the first PCR and nested inner primers. All other conditions remained the same. Primer sequences for the SuperScript IV Reverse Transcriptase synthesis and PCR amplification are shown in Table 1.

### 4.5. Gel Analysis and Imaging

RT-PCR amplicons were resolved on a 2% agarose gel in TAE buffer using a 100 bp DNA ladder (ThermoFisher Scientific, Melbourne, Australia) as the size standard. Exon skipping efficiency was analysed on a Fusion FX gel documentation system (Vilber Lourmat, Marne-la-Vallée, Collégien, France) for image acquisition and Bio-1D software (Vilber Lourmat, Marne-la-Vallée, Collégien, France) for analysis, as described previously [30,31]. All transfections and RT-PCR analyses were carried out at least twice.

### 4.6. Western Blotting

Western blotting was performed using a protocol derived from Cooper et al. 2003, Nicholson et al. 1989, 1992 and Aartsma-Rus et al. 2003 [12,32,33,34]. Cells were harvested and resuspended in treatment buffer (4.5 mg wet pellet weight/100 µL) consisting of 125 mmol/L Tris-HCl pH 6.8, 15% (*w*/*v*) sodium dodecyl sulfate, 10% glycerol (*v*/*v*), 0.5 mmol/L phenylmethylsulfonyl fluoride (PMSF) (Merck, Bayswater, Australia)), 50 mmol/L dithiothreitol, bromophenol blue (0.004% *w/v*) and a protease inhibitor cocktail (3 μL/100 μL of treatment buffer) (Merck, Bayswater, Australia). Samples were vortexed briefly, sonicated for 1 s, 4–8 times at a setting of 30/100 on an ultrasonic processor (Sonics, Newtown, CT) and heated at 95 °C for 5 min. Samples were first loaded onto a gel and myosin expression was assessed using Coomassie blue staining. Samples with equal loading according to myosin quantity, which will allow for quantitation by direct comparison thereafter, were then electrophoresed on a Biorad Criterion Bis Tris 4–12% gradient gel at 200 volts for one hour at room temperature. Fractionated proteins were transferred to an Immobilon-FL (Merck, Bayswater, Australia) PVDF membrane overnight at 18 °C at 290 mA, in a transfer buffer without methanol. Dystrophin was detected with NCL-DYS1 monoclonal anti-dystrophin (Novocastra, Newcastle upon Tyne, UK) applied at a dilution of 1:1000 for 2 h at room temperature. Detection was performed using a Western Breeze kit according to the manufacturer’s instructions (ThermoFisher Scientific, Melbourne, Australia). Enhanced chemi-luminescence was detected directly using a Fusion FX gel documentation system (Vilber Lourmat, Marne-la-Vallée, Collégien, France), using FusionCapt Advance software for image acquisition and Bio-1D software (Vilber Lourmat, Marne-la-Vallée, Collégien, France) for image analysis.

## Figures and Tables

**Figure 1 ijms-21-04511-f001:**
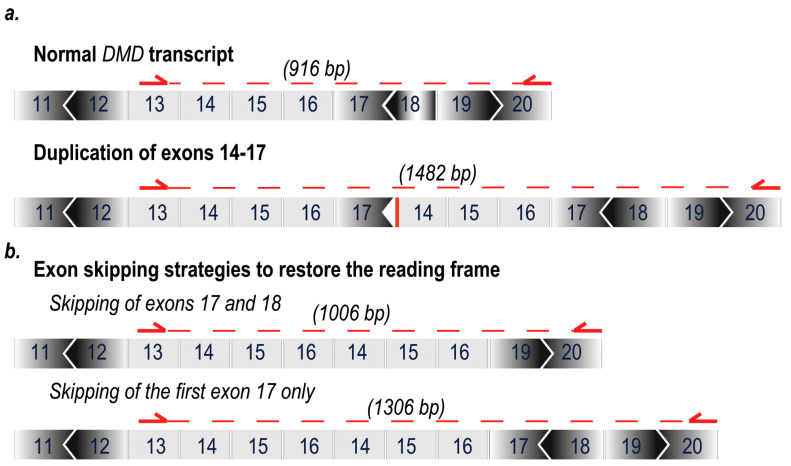
Antisense oligomer (AO) mediated exon excision strategies to restore dystrophin expression in the presence of frame-shifting exon duplications. AO strategies targeting exons 17 and 18 to generate a truncated yet in-frame dystrophin transcript. (**a**) Normal dystrophin exon arrangement and reading frame disrupted by a duplication of exons 14–17; (**b**) excision of exons 17 and 18 in a DMD transcript with an exon 14–17. Excision of only the first copy of exon 17 from dystrophic cells with an exon 14–17 duplication. Arrows and dotted line indicate predicted PCR amplicon sizes.

**Figure 2 ijms-21-04511-f002:**
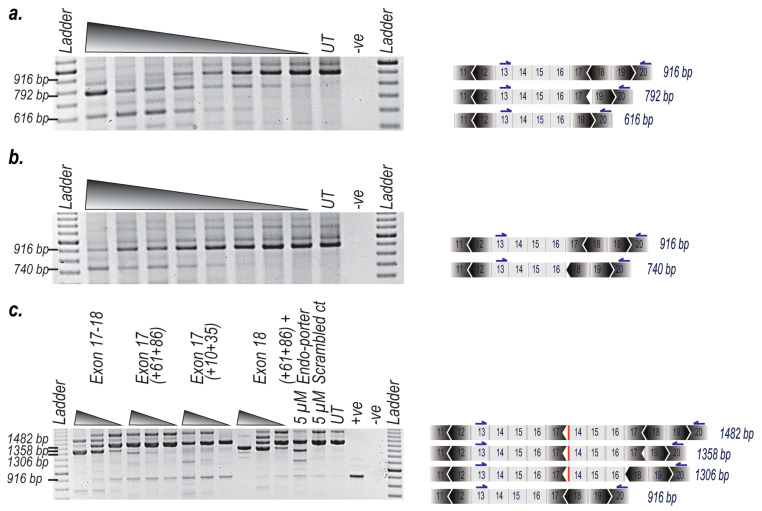
AO-induced excision of exons 17 and 18 from the *DMD* transcript in primary myogenic cells derived from a healthy donor and from a dystrophic patient with a duplication of exons 14–17, after transfection with phosphorodiamidate morpholino oligomer coupled to a cell penetrating peptide (PPMO) and/or phosphorodiamidate morpholino oligomer (PMO). Normal myogenic and dystrophic myogenic cells were transfected with PMO or PPMOs and incubated for 96 h. Total RNA was extracted, and RT-PCR was undertaken; (**a**) RT-PCR of RNA from normal myogenic cells transfected at concentrations of 5 µM, 1 µM, 800 nM, 400 nM, 200 nM, 100 nM, 50 nM and 25 nM with PMO and PPMO targeting exons 17 (H17A (+61+86)) and exon 18 (H18 (+24+53)) (RT-PCR was undertaken across exons 13–20. Transcript product sizes; full length is 916 bp, deletion of exon 18 is 792 bp and deletion of exon 17 and 18 is 616 bp); (**b**) RT-PCR of RNA from normal myogenic cells transfected at the concentrations of 5 µM, 1 µM, 800 nM, 400 nM, 200 nM, 100 nM, 50 nM and 25 nM with PMO targeting exon 17 (H17A (+61+86)) (RT-PCR was undertaken across exons 13–20. Transcript product sizes; full length is 916 bp and deletion of exon 17 is 740 bp); (**c**) RT-PCR of RNA from exon 14–17 duplication cells transfected at the concentrations of 5 µM, 800 nM and 200 nM with PMOs targeting exon 17 and/or exon 18 PPMO (RT-PCR was undertaken across exons 13-20. Transcript product sizes; full length is 1482 bp, deletion of exon 18 is 1358 bp, deletion of exon 17 is 1306 bp and the normal sized product is 916 bp); UT: amplicon from untreated exon duplication cells; +ve: amplicon from untreated normal cells; -ve: no template PCR negative control.

**Figure 3 ijms-21-04511-f003:**
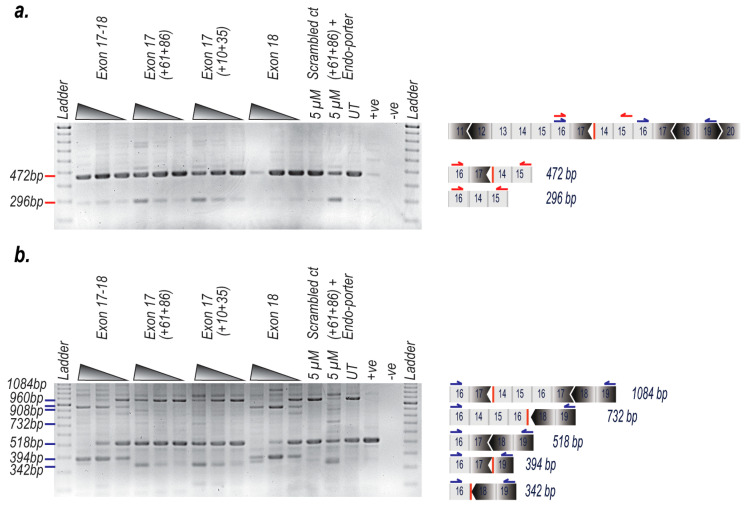
PCR analysis to determine specific AO-induced excision of dystrophin exons 17–18 from a dystrophic patient myogenic cell strain with a duplication of exons 14–17. Dystrophic myogenic cells were transfected with PMO or PPMOs at 5 µM, 800 nM and 200 nM and incubated for 96 h. Total RNA was extracted and RT-PCR was undertaken across exons 16-15 and 16-19 to analyse the proximal (5’) and distal (3’) ends of the duplication; (**a**) RT-PCR of RNA from exon 14–17 duplication cells transfected with the PMO targeting exon 17 and/or exon 18 PPMO using primers spanning the 5’ junction of the duplication (transcript product sizes; full length is 472 bp, deletion of exon 17 is 296 bp); (**b**) RT-PCR of RNA from exon 14–17 duplication cells transfected with PMO targeting exon 17 and/or exon 18 PPMO using primers spanning the 3′ end of duplication. The “full length” amplicons are either 1084 or 518 bp long, depending upon whether PCR was initiated at the first or second exon 16 (transcript product sizes; full length is 518 bp, deletion of exon 17 is 342 bp and deletion of exon 18 is 394bp); UT: amplicon from untreated exon duplication cells; +ve: amplicon from untreated normal cells; -ve: no template PCR negative control.

**Figure 4 ijms-21-04511-f004:**
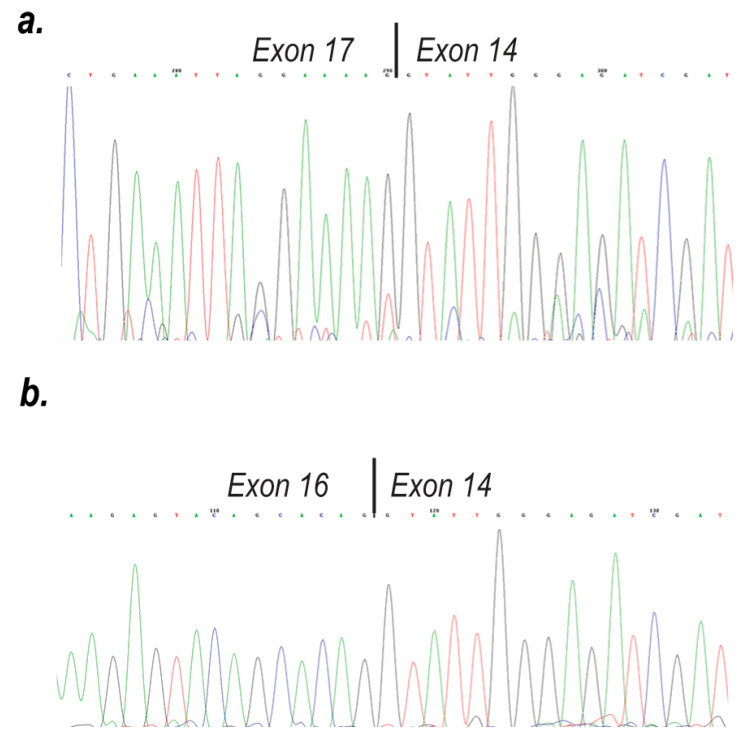
(**a**) Sequence chromatogram showing the duplication of exons 14–17, with exon 17 being spliced to exon 14 across the duplication junction; (**b**) Sequence chromatogram showing the removal of the first exon 17, with exon 16 being spliced to exon 14 across the duplication junction.

**Figure 5 ijms-21-04511-f005:**
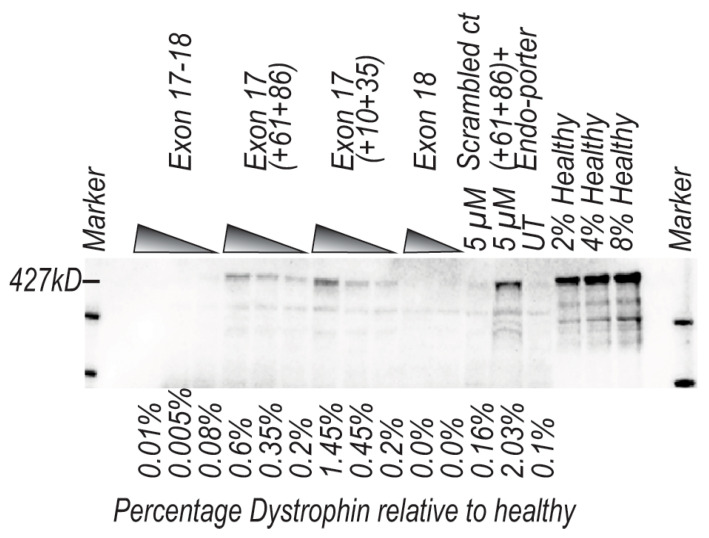
Western blotting analysis of AO-induced excision of dystrophin exons 17 and 18 from a dystrophic patient myogenic cell strain with a duplication of exons 14–17 after transfection with PMO and/or PPMO. Myogenic dystrophic cells were transfected with PMO and/or PPMO targeting exons 17 and 18 at 5 µM, 800 nM and 200 nM and incubated for 96 h. Protein was extracted, and western blotting was performed; western blot analysis showing the presence of DMD protein at 427kD. Protein loading was standardized according to myosin heavy chain densitometry performed on a Coomassie blue stained gradient gel. The quantity for each band was directly related to the healthy control.

**Figure 6 ijms-21-04511-f006:**
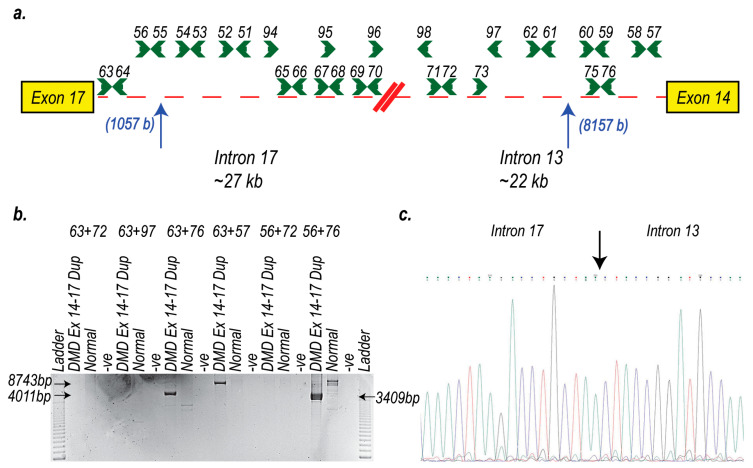
PCR analysis of the intron 17: intron 13 junction to map the breakpoint of a dystrophic patient myogenic cell strain with a duplication of exons 14–17. DNA was extracted from cells derived from a patient carrying a dystrophin exon 14–17 duplication and from myogenic cells from a healthy donor. PCRs were undertaken across intron 17 and 13 to analyse the breakpoint between the proximal exon 17 and distal exon 14. (**a**) Intron 13 and intron 17 junction showing primer locations across both introns; (**b**) PCRs showing products of 3409 bp (56+76), 4011 bp (63+76) and 8743 bp (63+57) present in the exon 14–17 duplication patient DNA but not in DNA from the healthy donor. -ve: no template PCR negative control; (**c**) Sequence chromatogram showing the junction of intron 17 and intron 13.

**Table 1 ijms-21-04511-t001:** Nucleotide sequences and annealing coordinates of AOs designed to remove dystrophin exons 17 and 18.

Nomenclature	Sequence (5′-3′)
H17A(+10+35)	AGU GAU GGC UGA GUG GUG GUG ACA GC
H17A(+61+86)	UGU UCC CUU GUG GUC ACC GUA GUU AC
H18A(+24+53)	CAG CUU CUG AGC GAG UAA UCC AGC UGU GAA
Scrambled Sequence	GGA UGU CCU GAG UCU AGA CCC UCC G

**Table 2 ijms-21-04511-t002:** Primer Sequences for cDNA synthesis and PCR.

Primer Location	Primer Sequence
Exon 13 Forward	TGC TTC AAG AAG ATC TAG AAC AAG AAC
Exon 13 Forward	CAC GCA ACT GCT GCT TTG GAA G
Exon 16 Forward	GCA AAC TGT ATT CAC TCA AAC
Exon 15 Reverse	GTG AAT CTT GTT CAC TGC ATC
Exon 19 Reverse	ACG TTC CAC CAG GGC CTG AG
Exon 20 Reverse	GAT ACT CCA GCC AGT TAA GTC T
Exon 21 Reverse	GGC CAC AAA GTC TGC ATC CAG
Exon 25 Reverse	GTC TCA AGT CTC GAA GCA AAC
Intron 17 Forward	TGG TGT TTT CTG GGG TGA AA
Intron 17 Forward	CCA GGG TAG AGA AGT TTG AA
Intron 13 Reverse	ACC ACT ATA TAA AAT GGC TCC
Intron 13 Reverse	GAG GGA GGC AGA AAA CAA CC

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
