# Peer review of "Single Exon Skipping Can Address a Multi-Exon Duplication in the Dystrophin Gene"

_ijms, 2020, doi:10.3390/ijms21124511_

Round 1
Reviewer 1 Report
In this manuscript, the authors evaluate the efficiency of different antisense oligonucleotide-induced splice switching strategies to remove exons 17 and/or 18 from the DMD gene and restoring dystrophin in primary patient myoblasts derived from a DMD patient with a DMD duplication of exons 14-17. The topic of this article is interesting, also considering the emerging evidence of Dystrophin exon skipping as a therapeutic strategy for some DMD.
Paper is well written, and data are presented in a clear and logical manner. I have the following minor comments to this work:
Abstract
-Line 17: please insert the abbreviation “DMD” for Duchenne muscular Dystrophy and use this abbreviation when necessary
Introduction:
-Line 42: please insert a reference at the end of the sentence…”disrupt the protein coding sequence”.
-Line 62: there is no space after the word “advantages”
-Please check along the abstract and the text how Duchenne Muscular Dystrophy is written (sometimes you use Duchenne muscular Dystrophy)
Results
-Lines 89-91: Please provide the results obtained with ESE finder 3.0 (also as supplementary materials).
-Lines 101-103: The authors say: “The PMO H17A (+61+86) targeting exon 17 was found to be more efficient than H17A (+10+35), which was designed to anneal to motifs closer to the exon 17 acceptor splice site (Fig. 2a, b)”. However, Fig. 2a and 2b are not referred to this result (Fig2a RT-PCR of RNA from normal myogenic cells transfected….. with PMO and PPMO targeting exons 17 (H17A (+61+86)) and exon 18 (H18 (+24+53); Fig2b: RT-PCR of RNA from normal myogenic cells transfected….. with PMO targeting exon 17 (H17A (+61+86)). Perhaps this result is shown in Fig2c, please clarify.
-Line 99: is not clear to me when you use PMO or PPMO. In other word why did you choose PMO for H17A (+61+86) and PMMO for H18A (+24+53))? Did you test the efficiency of all possible combinations (e.g. PMMO for H17A (+61+86) and PMO for H18A (+24+53; ecc..))?...The same thing for the use of “Endo-porter TM”, why did you use this strategy only for PMO H17A (+61+86)?. Since the main aim of this work is to identify the most efficient strategy to induce exon skipping my suggestion is to better clarify this point.
-Figure 2. Please provide on the right of the figure 1a, 1b and 1c the scheme of the primer location and expected amplicon sizes for the PCR analysis (as done in the figure 3).
-Figure 5. How did you normalize the western blot data? Please provide more information in the figure and in the material and methods section. This is very important since some treatment seems to be toxic for the cells (lanes 140-141: “However, when targeting exon 18 with the PPMO we observed substantial cell death at 5 μM, making western blot analysis difficult”).
-Lanes 208-215: please better explain in the results and discussion sections why this PCR analysis was carried out.
Author Response
Response to Reviewers comments
Thank you to both reviewers for the constructive comments in a very timely manner. The authors appreciate this.
Reviewer 1
Abstract
Line 17: please insert the abbreviation “DMD” for Duchenne muscular Dystrophy and use this abbreviation when necessary
Line 17- Abbreviation DMD has been included
Introduction
-Line 42: please insert a reference at the end of the sentence…”disrupt the protein coding sequence”.
Line 43- Reference has been inserted (Koenig et al 1989)
Line 62: there is no space after the word “advantages”
Line 63- A space after the word advantage has been added
-Please check along the abstract and the text how Duchenne Muscular Dystrophy is written (sometimes you use Duchenne muscular Dystrophy)
Duchenne muscular dystrophy has been corrected and is used consistently throughout the manuscript
Results
-Lines 89-91: Please provide the results obtained with ESE finder 3.0 (also as supplementary materials).
Lines 90-92- A previous paper by our laboratory with reference to ESE finder 3.0 predicting motifs in exon 17 and 18 has been included (Forrest et al, Figure 3).
Lines 101-103: The authors say: “The PMO H17A (+61+86) targeting exon 17 was found to be more efficient than H17A (+10+35), which was designed to anneal to motifs closer to the exon 17 acceptor splice site (Fig. 2a, b)”. However, Fig. 2a and 2b are not referred to this result (Fig2a RT-PCR of RNA from normal myogenic cells transfected….. with PMO and PPMO targeting exons 17 (H17A (+61+86)) and exon 18 (H18 (+24+53); Fig2b: RT-PCR of RNA from normal myogenic cells transfected….. with PMO targeting exon 17 (H17A (+61+86)). Perhaps this result is shown in Fig2c, please clarify.
Line 101-105- The authors thank the reviewer for bringing this to our attention. We have amended the text to reflect the correct PCR.As this is what we consider background information we have only shown the best PMO (H17A +61+86) in normal myogenic cells (Figure 2b). Figure 2a shows PMO H17A (+61+86) and PPMO H18A (+24+53) transfected together in equimolar amounts.
Line 99: is not clear to me when you use PMO or PPMO. In other word why did you choose PMO for H17A (+61+86) and PMMO for H18A (+24+53))? Did you test the efficiency of all possible combinations (e.g. PMMO for H17A (+61+86) and PMO for H18A (+24+53; ecc..))?...The same thing for the use of “Endo-porter TM”, why did you use this strategy only for PMO H17A (+61+86)?. Since the main aim of this work is to identify the most efficient strategy to induce exon skipping my suggestion is to better clarify this point.
Line 101-105- The authors understand the question regarding inconsistency of chemistry used and we have tried to make it easier to distinguish between the two in the text. The main reason is availability (we already had PMO for exon 17 and PPMO for 18) and although this is an interesting case, it is probably unique and the benefit of using PPMOs for exon 17 did not justify the increased cost. Regarding testing all combinations, all combinations were tested initially, and then the most promising used for protein studies and included in the manuscript. These dystrophic cells have limited myogenic capacity and eventually exhausted therefore we had to be careful with which treatments we could further study. Unfortunately, this is the case with many diseased cells.
Figure 2. Please provide on the right of the figure 1a, 1b and 1c the scheme of the primer location and expected amplicon sizes for the PCR analysis (as done in the figure 3).
Figure 2- Thank you for the suggestion, we have included the schematic showing primers and PCR products in figure 2
Figure 5. How did you normalize the western blot data? Please provide more information in the figure and in the material and methods section. This is very important since some treatment seems to be toxic for the cells (lanes 140-141: “However, when targeting exon 18 with the PPMO we observed substantial cell death at 5 μM, making western blot analysis difficult”).
Figure 5- We have amended the materials and methods section (western blotting lines 374-376) as well as the figure 5 legend (line 211-213). Samples were first loaded onto a PAGE gel and myosin was assessed by Coomassie blue staining and densitometry. Samples were then equally loaded on the gradient gel destined for western blot equally loaded according to myosin content (reflective of muscle protein rather than total protein), which allows for quantitation by direct comparison thereafter. The high concentration of the Exon 18 treatment was induced cell death, which was problematic, as the volume of the sample loaded was much higher than for the other samples.
-Lanes 208-215: please better explain in the results and discussion sections why this PCR analysis was carried out.
Line 215-216: We have included some additional text to explain why we performed the PCRs, in addition, this is also addressed in the discussion (lines 280-290).
Reviewer 2 Report
The manuscript by Greer et al, describes an exon skipping strategy to restore DMD gene product in a myogenic dystrophic cell strain where the exons 14-17 are duplicated. The manuscript is well written and the results clearly support the restoring of dystrophin reading frame by excision of only the first exon 17 in the DMD transcript carrying the duplicated 14-17 exons.
Authors should delete the line n. 377.
Author Response
Reviewer 2
Thank you for the comments
Line 377- please delete
Line 385 has been deleted